# Anatomic Investigation of Two Cases of Aberrant Right Subclavian Artery Syndrome, Including the Effects on External Vascular Dimensions

**DOI:** 10.3390/diagnostics10080592

**Published:** 2020-08-14

**Authors:** Mitchell H. Mirande, Madelyn R. Durhman, Heather F. Smith

**Affiliations:** 1Arizona College of Osteopathic Medicine, Midwestern University, Glendale, AZ 85308, USA; mdurhman96@midwestern.edu; 2Department of Anatomy, College of Graduate Studies, Midwestern University, Glendale, AZ 85308, USA; hsmith@midwestern.edu; 3School of Human Evolution and Social Change, Arizona State University, Tempe, AZ 85287, USA

**Keywords:** retroesophageal right subclavian artery, aberrant subclavian artery, arteria lusoria, dysphagia lusoria, kinked vertebral artery, artery tortuosity, Kommerell’s diverticulum

## Abstract

The retroesophageal aberrant right subclavian artery (ARSA) is a variation of the aortic arch that occurs asymptomatically in most patients. However, when symptomatic, it is most commonly associated with dysphagia. ARSA has also been noted as a location of potentially severe aneurysms in some patients, as well as posing a risk during surgical interventions in the esophageal region. This case study analyzes two individuals with ARSA morphology in comparison to a normal sample in order to gain a better anatomical understanding of this anomaly, potentially leading to better risk assessment of ARSA patients going forward. The diameter of the ARSA vessel was found to be substantially larger than both the right subclavian artery and brachiocephalic trunk of the subjects with classic aortic arch anatomy. As many ARSA individuals are asymptomatic, we hypothesize that the relative size of the ARSA may dictate its contribution to the presence and/or severity of associated symptomatology.

## 1. Introduction

The retroesophageal aberrant right subclavian artery (ARSA) or arteria lusoria is a vascular morphology that has been periodically reported in the literature and represents the most common aortic arch anomaly [1,2]. The ARSA represents about 17% of aortic malformations [1,3], has an incidence of 0.2–2.5% [1,4,5], a prevalence of 0.7–2.0% [6] and has a female predominance [6,7,8]. Clinical presentations of the ARSA include dysphagia (difficulty swallowing), dyspnea (labored breathing), chest pain, cough and weight loss [5]. Case studies have been presented discussing the discovery of this anatomic variant while uncovering the root cause of an individual patient’s symptoms, which if any, have been most commonly linked to dysphagia (difficulty swallowing) [1,2,7,8].

While ARSA often goes undiagnosed, certain imaging techniques can assist in its identification and diagnosis in patients with suspected symptoms. Barium contrast studies, computed tomography (CT) and magnetic resonance (MR) angiography have all demonstrated to be effective techniques for evaluating and diagnosing the ARSA condition [5,6,9,10,11]. In particular, CT angiograms, which carry the advantage of providing high spatial resolution imaging [6], can reveal the anomalous branching pattern and resulting vascular anomalies associated with ARSA [9]. Mahmodlou and colleagues described the importance of using CT to evaluate ARSA and other vascular anomalies prior to surgical interventions of the esophagus [5]. When an aberrant subclavian artery impinges on the esophagus causing dysphagia, barium contrast studies also may be a useful diagnostic tool [6,9,10].

The classic anatomic arrangement of the right subclavian artery is as a branch off the brachiocephalic trunk. However, in individuals with ARSA, the right subclavian artery branches directly off the posterior aortic arch (Figure 1) [1,3,4,5,7,8]. This morphology affects its position and course relative to the mediastinal structures. In particular, rather than coursing directly medially through the thorax, an ARSA typically takes a more posterior route, most often traveling posterior to the esophagus and trachea (Figure 1) [1,2,3]. This aberrant position may cause compression of the aforementioned structures, resulting in a range of clinical symptoms.

This case study analyzes a sample of cadavers including two cadaveric specimens with an ARSA from an empirical approach to help elucidate the anatomic development and clinical implications of the retroesophageal right subclavian variant.

## 2. Materials and Methods

### 2.1. Samples

The primary specimens in this case study were two cadavers with type G [3] retroesophageal right subclavian artery (ARSA) variants (1 M/1 F) from the anatomy teaching laboratory at Midwestern University. This unexpected finding in the teaching laboratory allowed for an opportunistic evaluation of the anatomy of this rare condition, including an analysis of external dimensions of the affected vasculature. In addition, twenty-three cadavers with classic aortic arch anatomy (11 M/12 F) were included in the study for comparative purposes. The causes of death were varied, and the subjects ranged in age from 63–95 at the time of death. Following student dissection of the cadavers, which consisted of removal of the lungs and heart, an initial assessment of the remaining aortic arch and branches was conducted. Further dissection was performed to open the visual field, allowing for arterial measurements to be collected. Cadavers with other abnormal aortic variants or loss of integrity of structures were excluded from consideration. This study was determined to be IRB-exempt by the Midwestern University Institutional Review Board (#AZ 1342).

### 2.2. Data Collection and Analysis

The primary goal of the present study was to evaluate the morphology of the two ARSA cadavers in comparison to the sample of non-ARSA cadavers. In particular, quantitative data on the external dimensions of vasculature that may have been impacted by ARSA were collected with Mitutoyo digital calipers. We investigated whether the right subclavian artery or its branches were larger in the ARSA cadavers. At each location, we measured the mediolateral width (*a* axis) and the anteroposterior width (*b* axis) at positions perpendicular to each other and calculated the surface area of each location using the equation for an ellipse (A = π
*ab*, a = radius of *a* axis width, *b* = radius of *b* axis width). The dimensions of the following arteries were measured: base of aorta, base of brachiocephalic trunk (if present), bases and midpoints of left and right subclavian arteries, bases and midpoints of left and right common carotid arteries, bases of right and left vertebral arteries, bases of the thyrocervical trunk and each of its branches. The measurement for the base of the aorta was collected three centimeters from its first branch, measured on the anterior surface. We also measured eight distances to illustrate the branching distances of the aortic and proximal vessels: base of right subclavian artery to its first branch (usually vertebral artery), base of right subclavian artery to its midpoint, base of right subclavian artery to base of brachiocephalic trunk (normal cadavers only), base of right common carotid artery to base of brachiocephalic trunk (normal cadavers only), base of right subclavian artery to base of right common carotid artery (variants only), base of right subclavian artery to base of left common carotid artery (variants only), base of left subclavian artery to its first branch (usually vertebral artery) and base of left subclavian artery to its midpoint. These values were compared between the cadavers with classic anatomy versus the ARSA variants. Potential sex differences were evaluated statistically using Analyses of Variance in SPSS 25 (IBM Corp.). Following the completion of all the measurements, a structural assessment was conducted for each of the cadavers and the qualitative findings were also noted.

### 2.3. Ethical Standard Statement

Cadavers utilized in this study were obtained from Midwestern University Body Donation Program in Glendale, AZ, USA. The dissection of cadaveric specimens was performed according to The Common Rule regulations established in the Code of Federal Regulations (USA). The Institutional Review Board at Midwestern University indicated that IRB approval was not required for this project (#AZ 1342).

## 3. Results

### 3.1. Comparison of Variants vs. Normal Subjects

The ARSA variant was identified in two cadavers, one of which was a 63-year-old white male who died of a malignant neoplasm of the liver and the other was a white 73-year-old female who died of ischemic cardiomyopathy. Both ARSA cadavers exhibited a right subclavian artery that was substantially larger than both the right subclavian artery and the brachiocephalic trunk of their normal subject counterparts (Figure 2). The base of the ARSA at the posterior surface of the aortic arch was enlarged (variant avg = 394.17 mm^2^ vs. normal avg. = 113.30 mm^2^) and subsequently decreased in size throughout its route posterior to the esophagus. However, it consistently remained larger than the right subclavian artery of the normal cadavers along its route to the upper right extremity. Neither ARSA variant presented with a Kommerell’s diverticulum (KOD) or had notable distal branching pattern anomalies. The bases of the aorta, right and left common carotid arteries and left subclavian artery were comparable in size between the two groups (Figure 2). Although there were no notable differences across the sample in the smaller downstream branches tested, the right thyrocervical trunk of the female ARSA cadaver #1 was the largest in the sample, measuring at about 10 mm larger than the mean of the non-ARSA cadavers (32.09 mm vs. 19.38 mm in non-ARSA).

### 3.2. Comparison of Sexes

Both female and male ARSA cadavers exhibited an enlarged right subclavian artery when compared to the average size of the normal cadavers; however, the disparity between the ARSA male and its normal counterparts was greater than the size disparity between the female variant and non-variant females (Appendix A and S2). In addition, the base of the aorta in the male variant was smaller compared to the average size of the normal cadavers, but the female variant base, although slightly larger, was more similar in size to the normal cadavers. In contrast, the female variant’s left subclavian artery was smaller than the average normal females, but the male variant was similarly sized to its normal counterparts. The right and left common carotid arteries in both ARSA variants were similar in size to the normal specimens, but it was noted that the female variant showed slightly more divergence than the male variant from the average of the normal cadavers. Across the sample, males had a significantly larger right common carotid artery than females (*p* = 0.041), but there was no significant difference in size of the right subclavian artery (*p* = 0.268).

## 4. Discussion

### 4.1. Size and Clinical Implications of Retroesophageal Aberrant Right Subclavian Artery

The findings in this study demonstrate that the ARSA is significantly enlarged compared to a normal right subclavian and brachiocephalic trunk as it branches off the posterior aspect of the aortic arch (Table 1 and Appendix A). Importantly, its distinctly large size has significant clinical implications. One of the most common symptoms of this variant is dysphagia [1,7,8], often referred to as dysphagia lusoria, which is due to compression of the esophagus as the aberrant right subclavian artery travels past the esophagus posteriorly to the upper right extremity. However, in most cases, the condition is asymptomatic (90–93% of cases [8]) [5]. It is currently unclear why some patients present with symptoms of dysphagia, dyspnea and chest pain [5], while many others are asymptomatic. Additionally, it is unknown whether ARSA patients are at a higher risk of more severe complications such as aneurysm [12]. We believe that the relative size of the ARSA may help dictate its contribution to the presence and/or severity of associated symptomatology, as discussed in further detail below.

Symptoms of the ARSA anomaly have been documented as presenting during the two extreme temporal periods of life due to anatomic changes [8,9]. In infants, the developing trachea is more flexible, which may allow the ARSA to compress the airway more easily, resulting in symptoms of stridor (wheezing caused by an obstructed airway) and respiratory tract infections [2,6,8]. A study conducted in 1993 found that 86% of infant patients with the ARSA morphology presented with stridor or recurrent respiratory infections [8,13]. On the other end of the temporal spectrum, in adults, respiratory symptoms co-occur with ARSA less frequently, as the adult trachea is more rigid. Instead, an inflexible trachea can lead to easier compression of the esophagus, which may contribute to the presentation of dysphagia as a primary symptom [8,13,14]. Other explanations of the presentation of dysphagia in older adults may result from the formation of an aneurysm, elongation of the aorta and increased rigidity of the esophagus or the vessel wall [3,8,15].

### 4.2. Coexisting Malformations of the Aberrant Right Subclavian Artery

An early study on ARSA morphology questioned whether the ARSA anomaly alone is sufficient to cause symptoms of dysphagia. Instead, it was argued that the presence of a coexisting bicarotid trunk (BCT), observed in 29% of their cases (85 out of 295 cases), was the cause for symptom presentation [3,8,16]. Subsequent studies have confirmed that the two conditions may co-occur [1,3,9,17]. However, a more recent study showed that while 71.2% of their 141 ARSA cases reported dysphagia as a symptom, only 19.2% (27/141) possessed the bicarotid morphology [8]. In addition, in many cases of a BCT co-occurring with an ARSA, barium-contrast examination indicates that the esophagus is compressed posteriorly and obliquely at the level of the aorta [3,4,6,9,15,18,19,20], suggesting that the ARSA is the primary contributor to esophageal compression and resulting dysphagia. There are also documented cases in which a BCT is absent or extremely mild, but dysphagia still presents [19,21]. These observations suggest that symptomology linked to the ARSA morphology is complex and multifactorial, with a variety of risk factors playing a role in the presentation of dysphagia or its variety of symptoms [6]. We propose that the larger that the diameter of the ARSA is, as it travels along its abnormal route posterior to the esophagus and trachea, the more likely it is to contribute to resulting compression and symptoms. Furthermore, the expansion of the lumen of the ARSA may result in compromise to the structural integrity of the vessel walls making aneurysms in this location more likely, causing more severe complications in these patients.

In addition to BCT, there are a variety of other vascular malformations that may coexist with the ARSA. It has been reported that the ARSA presentation may occur in Edwards (55%), Down’s (7.9–29.6% in fetuses and 1.6–35.7% in adults), Turner (43%), Patau (50%), DiGeorge (14%), Potter, post-rubella and Noonan syndromes [7]. Outside of these syndromes, the ARSA may coexist with other cardiovascular malformations such as the tetralogy of Fallot (12%) [1] and can increase the likelihood of co-occurring visceral anomalies such as asplenia, gall bladder agenesis, esophageal atresia, trachea–esophageal fistula, anal atresia, lung lobation abnormalities, double uterus and vagina, renal anomalies and sacral spina bifida [7].

### 4.3. Aneurysm of the Aberrant Right Subclavian Artery

Aneurysms associated with the ARSA anomaly, although rare [12,22], are a severe complication when present. Understanding the risk that an ARSA patient has of developing an aneurysm would help clinicians better weigh the treatment options when caring for an ARSA patient. From our study, we determined that the ARSA has a drastically larger diameter at its branching point from the aorta than the corresponding arteries in normal individuals. We hypothesize that this enlarged vessel is at a higher risk of aneurysm formation due to a loss of vessel wall integrity during development. Additionally, this risk may increase with the presence of a Kommerell’s diverticulum (KOD), which are more likely to develop in ARSA patients, due to aneurysmal degeneration [10]. The KOD, which is a diverticulum at the proximal descending aorta that gives rise to an aberrant subclavian artery in both the left and right aortic arch configurations, is a rare anomaly [23], which has been reported to occur in about 20–60% of individuals with an ARSA morphology [6]. If a Kommerell’s diverticulum progresses to an aneurysm, it can carry a higher risk of mortality [2]. The true rupture or dissection rate of an aneurysm associated with a KOD is quite variable ranging from 0–50% [6]. Although more specific studies have reported first encounter rupture rates of 6% (2/33 cases) [24] and 19% (6/32 cases) [2,6] as well as dissection and/or rupture rates of 44% [24] or a sole dissection rate of 11% [6].

The subclavian artery is also characterized as a muscular type artery, which has a low percentage of elastic fibers in the tunica media [7], which may further explain the risk of aneurysm near the subclavian artery’s branch point in ARSA patients [7]. Additionally, a study conducted by Kim et al. reported histologically the presence of cystic medial degeneration, which is the most common finding in a surgically resected specimen from patients with an aortic aneurysm or dissection, in the vascular wall of patients with a KOD [Kim]. The patients in this study had ARSA vessels which ranged in size from 6–10 mm and were accompanied by KODs which ranged from 15–45 mm [23]. The two cadaveric specimens in our study presented with ARSA vessels with a size of 19.18 mm (F) and 25.26 mm (M) with no associated KODs. These findings show that the presence of a KOD is a risk factor for an aneurysm or dissection to present. However, there are also large ARSA vessels that develop in the absence of a KOD which may not present with the same aneurysmal risk but result in a set of risks and symptoms due to their distinct morphology.

### 4.4. Embryological Origin of the Aberrant Right Subclavian Artery

During development, the right subclavian artery (RSA) derives from two sources. The proximal RSA develops from the right fourth aortic arch, whereas the distal portion of the RSA develops from the right seventh intersegmental artery on the dorsal aorta [2,7,25]. The ARSA variant morphology develops when the right fourth aortic arch fails to develop, which results in the right seventh intersegmental artery remaining attached to the dorsal aorta, which derives the ARSA formation [2,7,8]. In approximately 80% of cases, the ARSA takes a retroesophageal course, with an interesophageotracheal course occurring in 16.7% of cases and 5% of cases traveling anterior to the trachea [7,18].

### 4.5. Diagnosis for the Aberrant Right Subclavian Artery

Discovering a patient’s ARSA anomaly is usually achieved incidentally [2] while investigating symptoms that may be comorbid among gastroesophageal reflux disease (GERD), respiratory disease and the ARSA variant. The use of barium contrast studies, computed tomography (CT) and magnetic resonance (MR) angiography have shown to be useful techniques for evaluating and diagnosing the ARSA anomaly.

#### 4.5.1. Barium Contrast Study

The use of a barium contrast study is an effective way to identify posterior compression of the esophagus by the ARSA vessel [9,10,11]. Multiple studies have described the incidental discovery of the ARSA vessel while performing a barium contrast study on patients presenting with dysphagia [6,9,10]. This method can help identify the location of posterior compression as well as determine the pulsation of the ARSA on the esophagus [9]. Typically, the oblique view of the barium study will reveal a diagonal indentation in the posterior surface of the esophagus [6]. Although the barium study can identify location and relative size of the ARSA, it cannot be used to determine the dimensions of the ARSA or its relationship to surrounding organs [6]. Further diagnostic methods should be used to perform a risk assessment and to develop a treatment plan for the ARSA variant once identified [9,10].

#### 4.5.2. Computed Tomography

Computed tomography (CT) is a useful tool to use in diagnosing the presence of an ARSA, although it has its disadvantages such as the use of iodine contrast media and irradiation [6,7]. The advantages of CT include its short scanning time, high spatial resolution imaging and availability, which can help in identifying the dimensions of the vessel, including the presence of a KOD [6,9,10,11]. Additionally, the use of a multidetector computed tomography (MDCT) scanner, with its three-dimensional images, can assist in the understanding of the patient’s unique anatomy and assist providers during decisions regarding treatment strategy [6]. For example, in a case report by Mahmodlou, the use of CT revealed an ARSA vessel in a patient prior to a procedure on an esophageal cancer, which allowed surgeons to alter their surgical procedure to prevent potentially life-threatening injury to the ARSA [5].

#### 4.5.3. Magnetic Resonance Imaging and Angiography

Magnetic resonance imaging (MRI) and magnetic resonance angiography (MRA), along with CT, are one of the current best diagnostic tools to use to identify the ARSA variant and its associated vascular anomalies such as a KOD [6,7,11]. MRI allows for visualization of any imaging plane and can identify the presence of esophageal compression and/or pulsation from the ARSA [6]. However, the use of MRI comes with a key disadvantage of requiring a greater length of time, up to 20–40 min for an examination [6]. The ARSA anomaly may also present with difficulty during MRA for angiographists or interventional cardiologists who utilize the brachial or radial approaches to reach the ascending aorta [1].

Although our cadaveric study does not contain imaging of these diagnostic techniques, various studies have shown the efficacy of these methods in identifying the ARSA morphology in living patients [5,9,10,11]. It is important for clinicians to be aware of the ARSA variant when using these diagnostic tools in relation to symptoms of dysphagia, dyspnea and chest pain [5,6].

### 4.6. Treatment for the Aberrant Right Subclavian Artery

Many patients with the ARSA morphology are asymptomatic and clinical intervention is not required [5]. When necessary, both medical and surgical treatments were described to treat the associated dysphagia, depending on the severity of the symptoms, variant anatomy and treatment response [18].

#### 4.6.1. Surgical Treatments

Patients presenting with symptoms related to compression of the esophagus or trachea, and those with aneurysmal dilation or a KOD may require surgical intervention [1,4,5,6]. A key issue in deciding on treatment type depends on whether an accompanying aneurysm is present [10]. If the ARSA is non-aneurysmal, then the goal of surgical treatment is the closure of the ARSA at its origin and the revascularization of the right subclavian artery via a transposition of the ARSA to the right common carotid artery [4,10].

Another issue that influences the decision to proceed with surgical treatment is whether or not a KOD is present. Indications for surgical treatment of a KOD in an asymptomatic patient remain disputed [6]. However, it has been stated that the decision of treatment strategy for a KOD should be based on the available surgical expertise and the anatomy and comorbidities of the patient [6].

A recent study (2020) described a less-invasive approach using a left thoracoscopic procedure and a right supraclavicular incision, which facilitates a safer closure of the ARSA and avoids dramatic displacement of the esophagus and trachea [4]. This approach also permits an ideal degree of exposure of the relevant neurovasculature as well as reducing postoperative morbidity [4].

If the ARSA is associated with an aneurysm, then surgical treatment will be required to address the aneurysm prior to the transposition of the ARSA [10], due to the high rate of rupture (22.6% [3,26]). The indication for surgical intervention of an aneurysm or an associated KOD has been described based on size by many investigators, although the details of the measurement location and method has not been well reported or uniform [6]. A study conducted by Cina et al., in 2004 suggested that a diameter of 3 cm at the level of diverticulum orifice was the threshold for surgery in low-risk patients [6]. Additionally, a study conducted in 2014 by Vucemilo et al., stated that treatment is recommended if an ARSA patient has an aneurysm >3 cm in diameter, has a symptomatic or ruptured aneurysm or is symptomatic [2]. Due to the anatomical position of ARSA aneurysms, it is very difficult to treat them completely using only a single surgical approach (supraclavicular approach, median sternotomy or left thoracotomy) [22]. The use of left thoracotomy, median sternotomy or bilateral carotid—subclavian bypasses followed by a thoracic aortic endograft have been indicated for patients with the presence of an ARSA aneurysm [4]. In patients with a symptomatic ARSA, with or without an associated KOD, a systematic review conducted by Vucemilo et al. showed that the hybrid open and endovascular surgical approach is a safe and effective method of treatment, reporting resolution of thrombosis and presenting symptoms while decreasing aneurysmal sac size, length of hospital stay and complications [2].

#### 4.6.2. Nonsurgical Treatments

For patients with less severe symptoms or those not wishing to move forward with surgical interventions, conservative management can be utilized which typically involves dietary modification as well as the use of proton pump inhibitors and prokinetic agents [11]. In a case series performed on six patients with dysphagia lusoria, conservative treatment with a proton pump inhibitor and a prokinetic agent showed to be 50% effective in treating symptoms [18].

### 4.7. Limitations

We concede that our analysis of only two ARSA specimens in this study limits our ability to make definitive claims about the significance of this morphology. However, compiling these data could assist future researchers in subsequent analyses and contribute to the understanding of this aortic anomaly. It is also possible that the embalming process or act of dissecting the cadavers may have altered the original positions of the structures studied; however, we consider preservation unlikely to have had dramatic effects on the measured dimensions. Further studies performed on in vivo subjects using CT angiograms would be the next step in furthering our functional comprehension of this variant.

## 5. Conclusions

This case study analyzed a rare aortic arch anomaly and demonstrated that not only the positioning of the branch point of the right subclavian artery deviated from the normal anatomic location, but that the size of the vessel was greatly enlarged compared to normal size. This observation is the beginning of gaining a better understanding of this anomaly and its range of presentations and comorbidities. This information and data set will add to prior studies and facilitate further research to gain a better understanding of the relationships between the ARSA variant and its associated variants and the potential risks these patients may face.

## Figures and Tables

**Figure 1 diagnostics-10-00592-f001:**
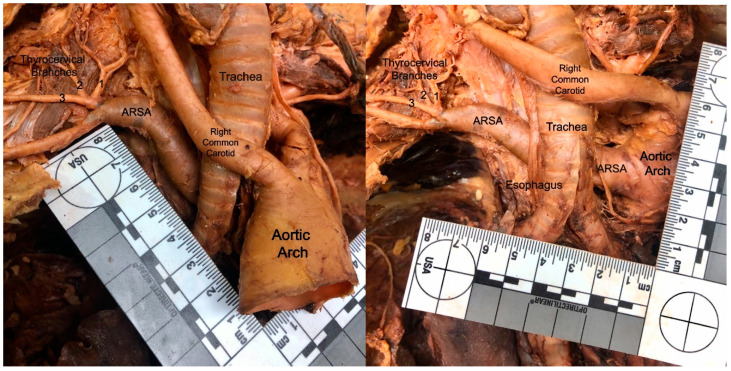
Dissection photo (left) showing the aberrant right subclavian artery (ARSA) of cadaver #1 (female), as it courses posterior to the esophagus and trachea. Dissection photo (right) shows the branching point of the aberrant right subclavian artery (ARSA), of cadaver #1 (female), from the posterior surface of the aortic arch. The aortic arch was retracted laterally to the left to gain this view.

**Figure 2 diagnostics-10-00592-f002:**
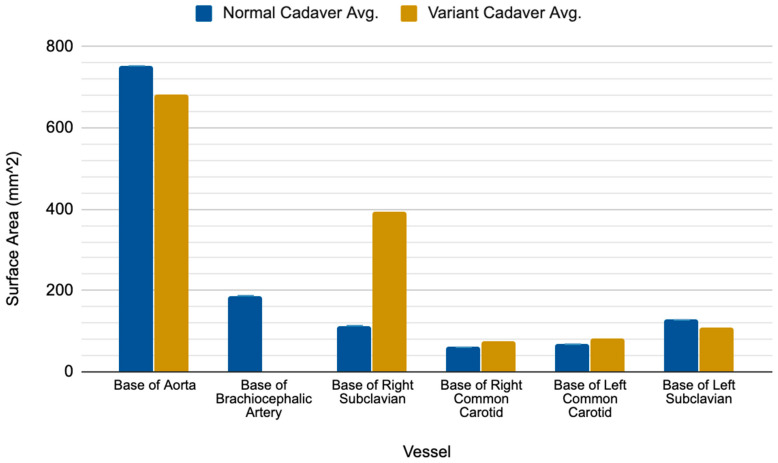
Comparative results showing the surface area measurements of aortic branches in the normal versus ARSA variant subjects. The surface areas at each location were averaged for each group.

**Table 1 diagnostics-10-00592-t001:** Cadaveric data comparing the surface area (SA) of a variety of vessels in both normal and variant subjects of both sexes.

Vessel	Normal Male Cadaver Avg. SA (mm^2^)	Variant Male Cadaver SA (mm^2^)	Normal Female Cadaver Avg. SA (mm^2^)	Variant Female Cadaver SA (mm^2^)
Base of aorta	829.80	642.68	675.99	721.30
Base of brachiocephalic artery	198.70	–	175.41	–
Base of right subclavian artery	122.08	482.69	104.52	305.65
Base of right common carotid	67.33	71.78	53.64	80.87
Base of right vertebral artery	25.39	15.45	19.40	–
Base of left subclavian artery	143.98	142.32	111.33	78.17
Base of left common carotid artery	72.87	84.48	62.84	79.61
Base of left vertebral artery	22.97	21.83	22.36	16.76

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
