# Peer review of "Anatomic Investigation of Two Cases of Aberrant Right Subclavian Artery Syndrome, Including the Effects on External Vascular Dimensions"

_diagnostics, 2020, doi:10.3390/diagnostics10080592_

Round 1

Reviewer 1 Report

Although the topic of this paper is very interesting, the current study is not well designed. Only two cadavers with an aberrant right subclavian artery (ARSA) were included.

Statistical analysis cannot be performed in such cases. Maybe the authors should redesign a new study with a larger sample of cadavers, as well as ARSA cases in order to safely conclude to severity of vertebral artery kinking and enlargment of other vessels' diameter.

Figures' depinction of kinking is not acceptable in my opinion. Why the authors only referred to the retroesophageal ARSA?

Other positions are not of interest? what type of study is the current investigation?

Maybe the authors should investigate separately the phenomenon of vertebral artery kinking.

Author Response

Thank you for taking the time to review our paper. We appreciate the helpful suggestions and comments that you offered and have taken steps towards addressing all of them below.

Although the topic of this paper is very interesting, the current study is not well designed. Only two cadavers with an aberrant right subclavian artery (ARSA) were included.

This article was intended to be presented as a case study, as we discovered this anatomical variant in two separate cadavers during dissection throughout the year. We have added in this descriptor into the study so that it is more clear that this isn’t a structured study design. Given the rarity of the ARSA variation (0.7-2.0% of the population) and its tendency to go undiagnosed, it is unfortunately not possible to design a cadaveric study with a large sample size of ARSA individuals. Each cadaver’s ARSA status is not known until after dissection, and thus cases are discovered opportunistically. In order to potentially obtain a sample of 30 ARSA cadavers, statistically speaking, we would have to dissect over 4285 cadavers, which is simply not feasible. Thus, we present our findings as a case report. 

Statistical analysis cannot be performed in such cases. Maybe the authors should redesign a new study with a larger sample of cadavers, as well as ARSA cases in order to safely conclude the severity of vertebral artery kinking and enlargement of other vessels' diameter.

Due to the adequate sample size of normal cadavers, we were able to perform statistical analysis on the normal cadavers in relation to each other when separated by gender. However, as correctly indicated, we were not able to run statistical analyses on the two ARSA variants as the sample size for variant individuals was too small. We have made this change within the paper (Lines 105-106, 117-123,147-148). 

Figures' depiction of kinking is not acceptable in my opinion. 

As suggested, we have removed the vertebral artery section completely from the manuscript (Lines 149-160, 248-273) and will save it for a separate study per your recommendation. 

Why did the authors only refer to the retroesophageal ARSA? Other positions are not of interest? 

In section 4.2. we discussed the other presentations of the ARSA in which the aberrant subclavian artery courses anterior to the trachea or in between the trachea and esophagus, although these conditions are much less common than the retroesophageal ARSA. The ARSA subjects in our study presented with the classic retroesophageal presentation, so we, therefore, focused our discussion on that morphology. As mentioned above, it would not be possible to knowingly obtain cadavers with alternate ARSA conditions, as the morphology is not revealed until after dissection. (Lines 282-284).

What type of study is the current investigation?

This is a case report based on the ARSA morphology discovered in two cadaveric subjects. We added descriptors into the paper to make this more clear. (Lines 18, 63, and 335).  

Maybe the authors should investigate separately the phenomenon of vertebral artery kinking.

As suggested, we have removed the vertebral artery section and will use it in a separate study where further investigation can be done. (Lines 149-160, 248-273).

Reviewer 2 Report

Comments: 

Dear Editor!

Thank you for possibility of reviewing the manuscript "Anatomical investigation of the roles of vascular dimensions and vertebral artery morphology in aberrant right subclavian artery syndrome". It is interesting manuscript focused on the correlation between dimension of some arteries and an aberrant right subclavian artery. Overall manuscript gives an appropriate report, but some recommendations should improve the paper.

  1. Abstract: change " arch anomaly" to "variation of the aortic arch"
  2. Data Collection and Analysis: add more clear definitions of measurements. Add description how did you calculate the surface area of each vessels?
  3. Results: add vertebral arteries (left and right) do analyzed arteries (results add also to table 1)
  4. Add abbreviation list
  5. Add to "Discussion" more detailed information describing coexistence of an aberrant right subclavian artery with another variations of aortic arch (suggested additional literature).
  6. Add limitation of the study

This manuscript is important from clinical point of view, so I recommend it for publication after revision.

Suggested additional literature:

  1. Association of aberrant subclavian arteries with aortic pathology and proposed classification system. Plotkin et al. J Vasc Surg. 2020 Mar 29:S0741-5214(20)30159-2
  2. Right aortic arch analysis - Anatomical variant or serious vascular defect? Arazińska et al. BMC Cardiovasc Disord. 2017 Apr 19;17(1):102.
  3. Aberrant Right Subclavian Artery with a Bicarotid Trunk: The Importance of Diagnosing This Rare Incidental Anomaly. Hanžič et al. Cureus. 2019 Nov 8;11(11):e6094

Author Response

Thank you for taking the time to review our paper. We appreciate the helpful suggestions and comments that you offered and have taken steps towards addressing all of them below.

Abstract: change " arch anomaly" to "variation of the aortic arch"

This change has been made. Kindly see the track changes document. (Lines 14-15).

Data Collection and Analysis: add more clear definitions of measurements. 

This change has been made. Kindly see the track changes document. (Lines 91-93, 96).

Add a description: how did you calculate the surface area of each vessel?

This change has been made. Kindly see the track changes document. (Lines 91-93). 

Results: add vertebral arteries (left and right) to analyzed arteries (results add also to table 1)

We have added both vertebral arteries into the description of the methods and added their results to Table 1. (Line 147-148).

Add abbreviation list

This change has been made. Kindly see the track changes document. (Lines 31-32).

Add to "Discussion" more detailed information describing the coexistence of an aberrant right subclavian artery with other variations of the aortic arch (suggested additional literature).

More information has been added to the discussion section in reference to other variations that occur alongside the aberrant right subclavian artery. Kindly see the tracked changes. (Lines 205-211).

Add limitation of the study

A section on limitations was written but inadvertently deleted from the manuscript during the second and final draft revisions. A limitations section has been added back into the manuscript. Kindly see the tracked changes document. (Lines 324-333).

Reviewer 3 Report

My questions and recommendations to the authors are as followed:

1)What is the postulated pathophysiology of aberrant subclavian artery enlargement (or aneurysm) formation?

2)On clinical grounds, one who does radial artery access for Endovascular intervention (whether it be cardiac or peripheral), need to be aware of the anatomy as it may cause unforeseen complications peri-procedure. Authors may want to add this in the manuscript.

Author Response

Thank you for taking the time to review our paper. We appreciate the helpful suggestions and comments that you offered and have taken steps towards addressing all of them below.

1) What is the postulated pathophysiology of aberrant subclavian artery enlargement (or aneurysm) formation?

In the fifth paragraph of section 4.1. we discuss the risk of aneurysm formation. In the literature, it was described that the vessel type of the subclavian artery, muscular type with a lower percentage of elastic fibers, puts the vessel at risk for the formation of aneurysms. Additionally, the presence of a Kommerell’s diverticulum (KOD) poses a risk due to the potential of it undergoing aneurysmal degeneration due to cystic medial degeneration which increases the risk of rupture if left untreated [10, 23]. We also recommended further studies be conducted on aberrant subclavian artery wall thickness and composition in patients with and without a KOD to determine if this additional data point could potentially point to an increased risk of aneurysms. (Lines 219-240).

2) On clinical grounds, one who does radial artery access for Endovascular intervention (whether it be cardiac or peripheral), need to be aware of the anatomy as it may cause unforeseen complications peri-procedure. Authors may want to add this in the manuscript.

This is an interesting point. However, given the focus of the paper on subclavian arterial morphology, we feel that clinical interventions on the radial artery would be beyond the scope of the paper. Thus, we respectfully decline to add a section on this topic to the paper.

Reviewer 4 Report

The clinical translation of the main findings into imaging techniques have to be improved. In accordance, the discussion needs to be changed in order to underline the clinical implications and enhance the significance of the manuscript. 

Author Response

Thank you for taking the time to review our paper. We appreciate the helpful suggestions and comments that you offered and have taken steps towards addressing all of them below.

The clinical translation of the main findings into imaging techniques have to be improved. In accordance, the discussion needs to be changed in order to underline the clinical implications and enhance the significance of the manuscript.

We thank the reviewer for mentioning this point. Unfortunately, imaging data were not available on our cadaveric subjects. Thus, it was not possible for us to include images or add results of that nature to the study. Therefore, we limit our study to direct cadaveric evidence that we were able to obtain. However, we appreciate the importance of imaging for future diagnosis and treatment and have therefore edited the Introduction and Discussion to include a more thorough discussion of these factors (Lines 44-53, 292-295).

Reviewer 5 Report

It was a pleasure to review this paper by Dr. Smith’s group. In this paper, the authors provide quantitative data regarding the dimensions of affected vasculature in patients with an aberrant right subclavian artery, as well as concomitant anomalies.

Major comments:

  • Although ARSA is the most common congenital abnormality of the aortic arch, clinical presentation is rare. In this study with only 2 cadavers, it is extremely challenging to draw any conclusions or statistical analysis from such a small sample size. I am curious on how the authors managed to extract statistical significant between the experimental and study groups.  
  • Results page 3 - it will be extremely helpful to see a demographics table for the patients with information such as their height, weight, past medical history etc. This is essential, as the size of vessels can directly correlate to the height and gender of patients. In this study, the authors compared and drew conclusions on the size of the vertebral and subclavian arteries, but without relating to patient characteristics.
  • Page 4 section 3.3: From my clinical experience and the available literature, I could not find any data that supports the following statement stated by the authors: “This anomaly has important clinical implications for the blood supply to the brain”. As long as the blood flow through the vertebral artery is anti-grade, kinking of the vertebral artery does not translate to a significant clinical relevance. This is because most patients have an intact circle of willis which centralizes blood flow to the brain. Therefore, the authors need to redefine the utility of characterizing “concomitant anomalies” in patients with ARSA (which was one of their objectives). If the authors want to support their statement, an ultrasound which demonstrates subclavian steel syndrome due to kinking of the vertebral artery needs to be added to this paper. However, this is challenging because we are dealing with cadavers only. Furthermore, while the authors did reference some papers that associate clinical symptoms of reduced posterior brain circulation and kinking of vertebral artery, I strongly believe that there are several confounding factors for such a statement and this needs to be clarified in the paper (perhaps in the discussion section).
  • There are key papers summarizing large case series which needs to be discussed to some extent in this paper (including but not limited to):
  • Tanaka Gen Thorac Cardiovasc Surg . 2015 May;63(5):245-59. doi: 10.1007/s11748-015-0521-3.
  • Cina J Vasc Surg. 2004 Jan;39(1):131-9. doi: 10.1016/j.jvs.2003.07.021.

Minor comments:

  • Introduction page 1 line 4- not all patients with ARSA are symptomatic
  • Introduction page 2 line 5 - There are some studies have studied the associated vascular dimensions in patients with ARSA. For instance, in a literature review by Vucemilo et al, the authors showed that over 50% of patients with ARSA had an associated thoracic aortic aneurysm. Thoracic aortic aneurysms can be deadly if they aren’t followed up and or managed.
  • The authors stated that the base of the ARSA at the posterior surface of the aortic arch was enlarged and subsequently decreased in size throughout its route posterior to the esophagus. This is expected in some patients with ARSA. However, it needs to be mentioned somewhere in the paper that patients with ARSA are at a high risk of developing Kommerell’s diverticulum “artery dilatation i.e aneurysm” at the base of the ARSA. This normally occurs due to the fact that the diverticulum is a remnant of the dorsal arch but has lost its continuity.
  • Discussion page 5, section 4.1 - The authors claim that the size of the ARSA causes dysphagia lusoria. However, it needs to be noted that it is not the size only but the anomaly location of the ARSA as well that are the main cause of dysphagia. Normally the subclavian is not behind the esophagus. Do we know if these two cadavers complained of dysphagia?

Author Response

Thank you for taking the time to review our paper. We appreciate the helpful suggestions and comments that you offered and have taken steps towards addressing all of them below.

Major comments:

Although ARSA is the most common congenital abnormality of the aortic arch, clinical presentation is rare. In this study with only 2 cadavers, it is extremely challenging to draw any conclusions or statistical analysis from such a small sample size. I am curious on how the authors managed to extract statistical significant between the experimental and study groups.  

Given the small sample size, we have removed the statistical analyses comparing ARSA and non-ARSA individuals from the paper. Instead, we now focus on descriptive differences in morphology between the groups and save our statistical analyses for the sex comparisons. Kindly see the tracked changes (Lines 105-106, 117-123,147-148). 

Results page 3 - it will be extremely helpful to see a demographics table for the patients with information such as their height, weight, past medical history etc. This is essential, as the size of vessels can directly correlate to the height and gender of patients. In this study, the authors compared and drew conclusions on the size of the vertebral and subclavian arteries, but without relating to patient characteristics.

We agree this information can provide useful insight; however, unfortunately, due to privacy reasons, there is limited demographic and medical information available on the donors. We have added the limited information available (sex, ethnicity, age at death, and cause of death) into the Results (Lines 111-113). 

Page 4 section 3.3: From my clinical experience and the available literature, I could not find any data that supports the following statement stated by the authors: “This anomaly has important clinical implications for the blood supply to the brain”. As long as the blood flow through the vertebral artery is anti-grade, kinking of the vertebral artery does not translate to a significant clinical relevance. This is because most patients have an intact circle of willis which centralizes blood flow to the brain. Therefore, the authors need to redefine the utility of characterizing “concomitant anomalies” in patients with ARSA (which was one of their objectives). If the authors want to support their statement, an ultrasound which demonstrates subclavian steel syndrome due to kinking of the vertebral artery needs to be added to this paper. However, this is challenging because we are dealing with cadavers only. Furthermore, while the authors did reference some papers that associate clinical symptoms of reduced posterior brain circulation and kinking of vertebral artery, I strongly believe that there are several confounding factors for such a statement and this needs to be clarified in the paper (perhaps in the discussion section).

As noted above, we have completely removed the section on the vertebral artery from the manuscript (Lines 149-160, 248-273), and will save it for a separate study where we are able to better explore this presentation. Thank you for your comments. 

There are key papers summarizing large case series which needs to be discussed to some extent in this paper (including but not limited to):

Tanaka Gen Thorac Cardiovasc Surg . 2015 May;63(5):245-59. doi: 10.1007/s11748-015-0521-3.

Cina J Vasc Surg. 2004 Jan;39(1):131-9. doi: 10.1016/j.jvs.2003.07.021.

We thank the reviewer for drawing our attention to these references. We have now read both of these papers and have added additional information to the paper. Kindly see the tracked changes (Lines 38-39, 47, 48, 53, 179, 195, 219-227). 

Minor comments:

Introduction page 1 line 4- not all patients with ARSA are symptomatic

In the abstract, introduction and discussion, we stated that this variant occurs asymptomatically in most patients but when present, is most commonly linked to dysphagia. Kindly see the track changes (Lines 14-15, 24, 170-171, 173, 296)

Introduction page 2 line 5 - There are some studies have studied the associated vascular dimensions in patients with ARSA. For instance, in a literature review by Vucemilo et al, the authors showed that over 50% of patients with ARSA had an associated thoracic aortic aneurysm. Thoracic aortic aneurysms can be deadly if they aren’t followed up and or managed.

After reading the suggested literature we have updated the paper with more information on both Kommerrell’s diverticulum and the risks of aneurysms.  Kindly see the tracked changes. (Lines 219-240).

The authors stated that the base of the ARSA at the posterior surface of the aortic arch was enlarged and subsequently decreased in size throughout its route posterior to the esophagus. This is expected in some patients with ARSA. However, it needs to be mentioned somewhere in the paper that patients with ARSA are at a high risk of developing Kommerell’s diverticulum “artery dilatation i.e aneurysm” at the base of the ARSA. This normally occurs due to the fact that the diverticulum is a remnant of the dorsal arch but has lost its continuity.

In section 4.1. we mentioned Kommerrel’s diverticulum in relation to aneurysmal degeneration and the increased risk of aneurysm formation in ARSA patients. We added to that paragraph more information clarifying the Kommerrel’s diverticulum relation to the ARSA and the risk it poses. Kindly see the tracked changes. (Lines 219-240)

Discussion page 5, section 4.1 - The authors claim that the size of the ARSA causes dysphagia lusoria. However, it needs to be noted that it is not the size only but the anomaly location of the ARSA as well that are the main cause of dysphagia. Normally the subclavian is not behind the esophagus. Do we know if these two cadavers complained of dysphagia?

We have added to that section language that makes it clear that the route posterior to the esophagus is abnormal and also a contributor to dysphagia.  (Lines 56, 58, 170, 201)

Due to privacy reasons, there is limited demographic and medical information available on the donors. We have added the limited information available (sex, ethnicity, age at death, and cause of death) into the Results (Lines 111-113).  Kindly see the tracked changes. 

Round 2

Reviewer 1 Report

Whereas the performed changes by the authors, the paper did not improved significantly. There is no connection in between the title of the paper and the conclusion of the manuscript. The novelty of the study is missing. A major revision is needed in order to highlight the role of vascular dimensions in aberrant right subclavian artery syndrome. Step by step the authors should give the design of the current paper. 

Author Response

Whereas the performed changes by the authors, the paper did not improved significantly. There is no connection in between the title of the paper and the conclusion of the manuscript. 

We thank the reviewer for evaluating the paper for a second time. We have now changed the title of the paper to better reflect the focus of the study. 

The novelty of the study is missing. A major revision is needed in order to highlight the role of vascular dimensions in aberrant right subclavian artery syndrome. Step by step the authors should give the design of the current paper.

This study was conceived opportunistically when two separate cadavers in the teaching laboratory at Midwestern University were found to possess the rare ARSA condition. This unexpected finding presented a unique opportunity to evaluate the anatomy of this condition, especially the external dimensions of the affected vasculature, which were observed to differ dramatically from the normal comparative sample. We have reworded the Materials & Methods to clarify the study design as an unexpected opportunity to perform a detailed case study and uncover previously unrecognized anatomical information (Lines 20, 73-80, 91-95)

We concede that this study, as with any such case study, has limitations for drawing broader conclusions, as we noted in the Limitations section of the Discussion (added in the first round of revisions). However, to our knowledge, this is the first study to evaluate how ARSA may affect the external dimensions of the right subclavian artery and its branches. We also respectfully note that none of the other four reviewers mentioned any concerns about the novelty of the study.

Reviewer 2 Report

Significant improvements were made.

Author Response

Significant improvements were made.

We thank the reviewer for their support.

Reviewer 5 Report

With pleasure I had the opportunity to review the resubmission by Dr.Smith’s group. In this version of the paper, the authors pivoted their manuscript into a case series with other modifications in the study design. Despite these changes, this case series is lacking a literature review, which most case series have. Given the current context of the paper, there is minimal contribution to the scientific literature. I recommend the authors work on increasing their sample size and complement their study with an up-to-date literature review.

Author Response

With pleasure I had the opportunity to review the resubmission by Dr.Smith’s group. In this version of the paper, the authors pivoted their manuscript into a case series with other modifications in the study design. Despite these changes, this case series is lacking a literature review, which most case series have. Given the current context of the paper, there is minimal contribution to the scientific literature. I recommend the authors work on increasing their sample size and complement their study with an up-to-date literature review.

We thank the reviewer for evaluating the paper for a second time. To clarify, the study is now organized as a case report, rather than a case series as stated by the reviewer. The study focuses on two subjects with the rare ARSA condition and evaluates their morphology relative to a large sample of anatomically normal individuals. We have clarified this focus in the Materials & Methods section of the manuscript (Lines 73-80, 91-95). 

We have expanded the literature review component of our study. In the last round of revisions, we added 3 citations and 36 lines of text to our literature review. In this round of revisions, we have added another 37 lines. However, since this study is not intended to be a case series, we do not feel it necessary to review all of the tangentially relevant literature. Instead, we focus on the papers most relevant to the ARSA cases identified here. (Lines 258-293, 315-319, 328-330)

As mentioned in our first round of revisions, given the rarity of the ARSA variation (0.7-2.0% of the population) and its tendency to go undiagnosed, it is unfortunately not possible to design a cadaveric study with a large sample size of ARSA individuals. Each cadaver’s ARSA status is not known until after dissection, and thus cases are discovered opportunistically. In order to potentially obtain a sample of 30 ARSA cadavers, statistically speaking, we would have to dissect over 4285 cadavers, which is simply not feasible. Thus, we present our findings as a case report.

Round 3

Reviewer 1 Report

The authors followed the reviewers' suggestions and made when possible appropriate corrections. The paper has improved. A final remark about the aberrancy of the vessel (ectopic origin and course) and its dimensions. Larger dimensions are related only with its origin or with its entire course? Branching pattern of ARSA was affected? When ARSA was considered aneurismal? Ectopia of a vessel is followed by an aneurysm creation? The authors should cite literature data and comment on possible origin of ARSA as penultimate branch. 

Author Response

The authors followed the reviewers' suggestions and made when possible appropriate corrections. The paper has improved.

We thank the reviewer for their support. 

A final remark about the aberrancy of the vessel (ectopic origin and course) and its dimensions. Larger dimensions are related only with its origin or with its entire course? 

The larger dimensions of the vessel are related primarily to the origin of the ARSA. However, we observed that it steadily decreased in size as it traveled out to the upper right extremity (URE) in both cases, and remained slightly larger than the average normal cadaver dimensions. Our ARSA subjects did not have a Kommerrell’s diverticulum which would present with an enlarged origin but more normal dimensions during its route to the URE. Kindly see the track changes document (Lines 117-118). 

Branching pattern of ARSA was affected? 

No further distal branching anomalies were observed in the ARSA individuals. This information has been added to the Results. Kindly see the track changes document (Lines 119-120).

When ARSA was considered aneurismal? 

Typically, the threshold for aneurismal intervention of an ARSA is considered to be a diameter of 3 cm or greater [2, 6]. This information has been added to the Discussion. Kindly see the track changes document (Lines 317-323).

Ectopia of a vessel is followed by an aneurysm creation? 

In terms of the ARSA, there are many factors that contribute to the formation of an aneurysm. In our paper, we have described a few mechanisms that have been proposed as to why an ARSA-associated aneurysm is most likely to form (Lines 207-210, 218-223). The ectopic nature of the ARSA on its own does not necessarily dictate aneurysm creation, although the combination of its ectopia with the factors referenced above may lead to aneurysm presentation.

The authors should cite literature data and comment on possible origin of ARSA as penultimate branch. 

The ARSA is most commonly reported to be branching off the posterior surface of the aortic arch [1, 3-5, 7, 8] and is therefore no longer in line with the other aortic vessels. Thus, it is difficult to conclude the branching position of the ARSA in relation to the other aortic vessels. Through our research, we found no mention of the ARSA branching within the area of the other aortic vessels.

Reviewer 5 Report

The authors addressed all of my comments.

Author Response

The authors addressed all of my comments.

We thank the reviewer for their support.